# Potential Use of Wearable Sensors to Assess Cumulative Kidney Trauma in Endurance Off-Road Running

**DOI:** 10.3390/jfmk5040093

**Published:** 2020-12-14

**Authors:** Daniel Rojas-Valverde, Rafael Timón, Braulio Sánchez-Ureña, José Pino-Ortega, Ismael Martínez-Guardado, Guillermo Olcina

**Affiliations:** 1Centro de Investigación y Diagnóstico en Salud y Deporte (CIDISAD), Escuela Ciencias del Movimiento Humano y Calidad de Vida (CIEMHCAVI), Universidad Nacional, Heredia 86-3000, Costa Rica; 2Grupo en Avances en el Entrenamiento Deportivo y Acondicionamiento Físico (GAEDAF), Facultad Ciencias del Deporte, Universidad de Extremadura, 10005 Cáceres, Spain; rtimon@unex.es (R.T.); wismi4@gmail.com (I.M.-G.); 3Programa Ciencias del Ejercicio y la Salud (PROCESA), Escuela Ciencias del Movimiento Humano y Calidad de Vida (CIEMHCAVI), Universidad Nacional, Heredia 86-3000, Costa Rica; brau09@hotmail.com; 4Departmento de Actividad Física y Deporte, Facultad Ciencias del Deporte, 30720 Murcia, Spain; josepinoortega@um.es

**Keywords:** renal health, wearable devices, technology, acute kidney injury, inertial measurement units (IMU)

## Abstract

(1) Background: This study aimed to explore wearable sensors′ potential use to assess cumulative mechanical kidney trauma during endurance off-road running. (2) Methods: 18 participants (38.78 ± 10.38 years, 73.24 ± 12.6 kg, 172.17 ± 9.48 cm) ran 36 k off-road race wearing a Magnetic, Angular Rate and Gravity (MARG) sensor attached to their lower back. Impacts in g forces were recorded throughout the race using the MARG sensor. Two blood samples were collected immediately pre- and post-race: serum creatinine (sCr) and albumin (sALB). (3) Results: Sixteen impact variables were grouped using principal component analysis in four different principal components (PC) that explained 90% of the total variance. The 4th PC predicted 24% of the percentage of change (∆%) of sCr and the 3rd PC predicted the ∆% of sALB by 23%. There were pre- and post-race large changes in sCr and sALB (*p* ≤ 0.01) and 33% of participants met acute kidney injury diagnosis criteria. (4) Conclusions: The data related to impacts could better explain the cumulative mechanical kidney trauma during mountain running, opening a new range of possibilities using technology to better understand how the number and magnitude of the g-forces involved in off-road running could potentially affect kidney function.

## 1. Introduction

Acute kidney injury (AKI) is a relatively uncommon condition in sports. This condition has been reported in prolonged and repetitive strenuous exercises [1]. It is understood as a transitional decrease in renal function, expressed by a reduction in glomerular filtration rate, increase in serum creatinine (sCr) and albumin (sALB), and alterations of other novel AKI-related urine and blood biomarkers during a relatively short period (1–3 days) [2].

The evidence of AKI cases in both contact and non-contact sports has been increased, but with clear different etiological backgrounds [3,4,5]. In contact sports like football, boxing, and rugby, AKI cases have been related to kidney contusion or trauma (grade I in American Association for Surgery of Trauma classification) during tackles, punches, or other high-intensity actions with direct impact to the body [6,7]. On the other hand, in non-contact sports (e.g., endurance running and cycling), AKI has been related to the high number of muscle eccentric-concentric contractions leading to muscle damage [8,9].

In endurance running and mainly off-road running [8,10], some evidence has been published regarding the impact of external workload (e.g., impacts) as an additional factor that may contribute to AKI incidence, next to other known factors like dehydration, heat strain, and high metabolic activity [11]. Within this multifactorial etiology, high physical internal and external load seems to be a discernible contributing factor to the transitory decrease in renal function in endurance runners [12]. It could be due to muscle damage in response to high eccentric actions and its effect on inflammatory and hemodynamic responses that may affect the kidney [13]. New evidence has also highlighted the cumulative mechanical trauma that affects the kidney during off-road running as a potential cause of AKI [9]. Although kidneys are very well protected structures, there is relative mobility that could lead to injury even when no direct trauma occurred [14], for example, during downhill running or change of directions during training or competition.

Monitoring physical load is critical in endurance sports, such as off-road running, due to the high number of actions involved [15]. This is why non-invasive tools as wearable sensors could be an accessible option to assess potential cumulative mechanical kidney trauma, indirectly analyzing the mobility of anatomical structures near the kidneys, such as the lower back. These wearable sensors are used to monitor physical load during exercise in different parts of the body, such as the wrist, waist, and trunk [16,17,18]. It has also been determined that there is a relationship between the increase in serum blood factors related to kidney damage and the quantified load in the lower back [9]. Therefore, this study aimed to explore the potential use of wearable Magnetic, Angular Rate and Gravity (MARG) sensors to assess cumulative mechanical kidney trauma during off-road running.

## 2. Materials and Methods

### 2.1. Design

Participants were asked to perform three loops of a 12 km (+ascend = 600 m) circuit (total distance = 36 km and total +ascend = 1800 m), under 25° Celsius of temperature, and 80% of humidity (Wet Bulb globe Temperature, 3M, USA). Runners wore a MARG sensor attached to the lower back during the race, and variables of time-related impacts were extracted. Two blood samples were collected pre- and post-race to assess serum creatinine (sCr) and albumin (sALB). An analysis was made to explore a model based on impact variables that explained sCr and sALB increases between pre- and post-race.

### 2.2. Participants

Eighteen experienced mountain runners participated in this study (age 38.78 ± 10.38 years, weight 73.24 ± 12.6 kg, height 172.17 ± 9.48 cm). They had 4.78 ± 2.42 years of experience competing in ultra-endurance events. Participant′s mean finish time was 4.2 ± 0.21 h. No neuromuscular, metabolic, or structural injuries were reported at least six months before the study. The participants were asked to avoid intense endurance exercise at least a week before the event.

All participants were notified of the study′s aim, protocol details and the potential risks and rights during their participation. The study´s protocol followed all biomedical guidelines based on the Declaration of Helsinki (2013) and it was reviewed and approved by the Institutional Review Boards of Universidad Nacional (Reg. Code 2019-P005) and Universidad de Extremadura (Reg. Code 139/2020).

### 2.3. Materials and Procedures

Sixteen different time-related impacts (*n*/min, g forces) variables were assessed using a Magnetic, Angular Rate and Gravity (MARG) sensor (WIMU PRO^TM^, RealTrack Systems, Almería, Spain). The devices were attached to the lower back (~L1–L3) [9] of each participant with a special spandex dark belt adjusted with elastic straps to avoid device´s unwanted vibrations or movements (see Figure 1). The MARG´s integrate four 3-axis microelectromechanical systems accelerometers (2x ± 16 g, 1x ± 32 g, and 1x ± 400 g), gyroscope, and magnetometer. All MARG´s calibration and setting were developed following published guidelines [19,20], its reliability for neuromuscular running physical load assessment has been proven [21] and its reliability has been tested in different body parts [22]. The variables extracted were total impacts per min (Impacts_Total_/min) and fifteen progressively scaled categories of g-force magnitude, each 1 g wide (Impacts_1–15 g_/min).

Blood serum samples were collected using 5 mL blood spray-coated silica tubes (BD Vacutainer^®^, Franklin Lakes, NJ, USA). After centrifugation (10 min at 2000 *g*), samples were stored at −20 °C. After 24 h, the samples were processed by the photometry method using an automatic biochemical analyzer (BS-200E, Mindray, China). The variable analyzed was serum creatinine (sCr, mg/dL) and serum albumin (sALB, IU/L). Acute kidney injury (sCr baseline in mg/dL *1.5) was considered following established diagnosis criteria [23]. Two groups were made based on AKI diagnosis as follows: those participants that met AKI diagnosis (AKI) and the ones that did not (No-AKI), in order to explore differences in the number of impacts reported.

Urine specific gravity (USG) was assessed as a hydration status marker. USG was confirmed and double-checked with a digital valid [24] handheld refractometer (Palm AbbeTM, Misco, Solon, OH, USA). It was classified following the hydration status ranges: well-hydrated <1.01, minimal dehydration 1.01–1.02, significant dehydration 1.02–1.03, and severe dehydration >1.03 [25]. The refractometer was cleaned with distilled water and calibrated previously. There were no reported urination problems or difficulties neither before nor after the race.

### 2.4. Statistical Analysis

All sixteen impact variables were grouped using a Principal Component Analysis (PCA) following previous studies guidelines [9,26]. PCA was suitable, according to Kaiser-Meyer-Olkin (*KMO* = 0.63) values and the Barleth Sphericity test (*p* < 0.01). Eigenvalues (EV) > 1 were considered for the extraction of each Principal Component (PC). A VariMax-orthogonal rotation method was used to identify the high correlation of components. A threshold of 0.6 was set to retain loadings. The highest loading was used when a cross-loading was found between PCs. PCA procedure followed standard quality criteria [27], meeting 21 out of 21 of the quality items.

A paired t-test was used to explore sCr and sALB changes between pre- and post-race data and the Change delta´s percentage (∆%) was calculated as follows: ((sCr post-race–sCr pre-race)/sCr pre-race)*100. An unpaired t-test was performed to explore potential differences in the number of impacts between those participants who met AKI diagnosis and those who did not. USG data were analyzed using a repeated measure *t*-test. The magnitude of the differences was calculated using Cohen´s *d*.

Finally, a stepwise regression model (*R^2^*) was applied to resulted factor scores obtained from impact´s PCA using the ∆% of sCr and sALB as the dependent variable. This statistical technique was applied to identify which impact´s PC could predict the ∆% of sCr, and ∆% of sALB.

All variables were presented in mean ± standard deviation. Alpha was set at *p* < 0.05 and all analyses were made using the Statistical Package for Social Science (v.22, SPSS, Chicago, IL, USA).

## 3. Results

Participants experienced a total of 170.57 ± 34.42 impacts per minute. Figure 2 shows the mean number of impacts per minute in relation to the associated magnitude of g-force (see Figure 2).

All sixteen impact-related variables were grouped in four different PC´s, explaining the 90.39% of total impacts cumulative variance. The 1st PC explained the 50.5% (EV = 8.08) of total variance, 2nd PC the 17.58% (EV = 2.81), 3rd PC the 13.05% (EV = 2.09), and 4th PC the 9.27% (EV = 1.48). Grouped variables and loadings are presented in Figure 3.

In follow up to the abovementioned PCA results, those participants that met AKI diagnosis criteria (33.3% of participants) registered lower number of impacts in the 1–2 g category (*t* = −2.42, *p* = 0.03, *d* = −1.45, large effect size) but higher number of impacts in the 14–15 g category (*t* = −3.1, *p* = 0.01, *d* = −1.58, large effect size) (see Figure 4.). No differences we found in the 5–6 g or 6–7 g categories.

There were large statistical differences (*t* = −6.24, *p* < 0.01, *d* = −1.47, large effect size) between sCr pre-race (1.24 ± 0.28 mg/dL) and sCr post-race (1.74 ± 0.41 mg/dL), and large differences (*t* = −2.78, *p* = 0.01, *d* = −1.47, large effect size) in sALB pre-race (4.33 ± 1.29 IU/L) vs. post-race (5.01 ± 0.86 IU/L). The ∆% of sCr was predicted by the 4th PC in a 24% (*R^2^* = 0.24, *β* = 44.03, *p* < 0.01) and the ∆% of sALB by a 23% (*R^2^* = 0.23, *β* = 100.55, *p* = 0.04). Finally, USG as a hydration marker reported no differences between pre- and post-race measurements (1.01 ± 0.02 vs. 1.01 ± 0.01; t = 1.02, *p* = 0.07).

## 4. Discussion

Renal injury provoked by an indirect trauma has been reported in previous cases with no symptoms other than lumbar pain but with radiological findings such as subcapsular renal hematoma [14]. Some evidence suggests that urinary trauma could be present in non-contact sports such as off-road running [4,5,28]. It has been hypothesized that kidney mechanical trauma could mediate in the development of acute kidney injury after running [9]. This could be due to the kidneys′ relative mobility during some actions as a downhill run at high speeds, change of directions, falls, and other high g-forces that could affect kidney movements and shaking. This relationship needs to be explored in future studies. The results of this study suggest that the 4th PC and 3rd PC of impact-related variables explained the ∆% of sCr and sALB between 23 to 24%. These findings indicate that the magnitude and number of impacts (g-forces) could have a potential role in the cumulative mechanical kidney trauma.

Despite kidneys being well protected by abdominal and back muscles, ribs, fat, renal pedicle, and ureteropelvic junction and supporting Gerota fascia in the retroperitoneum, they are also susceptible to internal movements [14,28]. Repeated sudden accelerations and decelerations may lead to renal contusions caused by the collision of kidneys in its surrounding tissues and structures like spine and ribs. These actions could lead to renal vasculatures affections, nephron damage, consequent hematuria, and other blood markers findings [29,30,31]. These accelerations and deceleration could be assessed using the variable impacts as proposed in this study. The impacts between 5–7 g explained the pre-post increase of sALB and the impacts of 1–2 g and 14–15 g explained the rise in sCr. Based on the literature [9], these results may suggest that both the volume and intensity of the impacts involved during renal contusions play a special role in acute kidney injury. It has been found that the ∆% of blood markers as serum creatine kinase and sCr could predict the external workload of wearable devices placed in L1–L3 by 40% and 27%, respectively [9]. This evidence supports the idea of a new hypothesis of mechanical kidney injury during endurance off-road running based on L1–L3 external workload data [9].

The results of the present study showed that MARG sensors could be used to register the impacts and g-forces that affect the lower back, which is the kidney´s nearest external structure of the body. MARG sensors could register vertical, anterior-posterior, and mediolateral forces using the integration of accelerometer, gyroscope, and magnetometer data. The g-forces provoked by sudden accelerations and decelerations may affect the kidneys. The number and magnitude of these impacts could be monitored using MARGs attached to the kidney´s nearest external structure of the body, the lower back. Kidneys typically extend from T12 to L3 and weigh 135–150 g, so the MARG positioning should be at this level despite a slight position change due to the kidney′s free mobility resulting from both body positions and respiration [32].

The link between the sensors′ external load and kidney trauma must be confirmed and discussed in future interventions. Previously, considering the cause of the increase in sCr may be indicative of kidney injury as well as massive muscle damage [33]. Although elevations in sCr in 33% of participants by itself should not be understood as kidney damage due to physical exercise, the rise in sALB could suggest transitory functional loss due to tubular or glomerular damage. In fact, there is evidence to suggest that proteins released into the bloodstream in high amounts (e.g., rhabdomyolysis) can overload kidney function, resulting in functional or subclinical damage reflected in an increase of sCr and sALB, respectively [34,35].

The cumulative small injuries during rough exercises as off-road mountain running might damage the kidney, resulting in AKI. Although there is no clear evidence that cumulative or subsequent AKI events contribute to future renal chronic conditions in athletes [1,36], there is enough evidence to suggest that athletes, coaches, and sports scientists should be concerned with controlling the kidney health of runners, monitoring those variables that can trigger AKI, and thus, preventing potential cases of this transitory kidney condition. Some preventive strategies have been proposed to endurance athletes such as optimal fluid and food intake, appropriate physical loading, rest, and acceptable recovery between efforts [3]. Monitoring physical load is essential and those external and internal variables that could affect not only kidney health but also general well-being should be assessed. Dehydration seems to be a factor that did not influence the AKI occurrence in this specific sample, as found in the results.

MARG units as wearable devices containing accelerometers, gyroscopes, and magnetometers allow trainers, athletes, and medical staff to monitor and control the physical external and internal loads involved during the off-road running. The information obtained would allow us to provide feedback on the kinematic behavior of the runner in an objective manner [37] and would facilitate the programming and prescription of training loads, preventing and mitigating the impact of AKI on the runner′s health and performance.

These findings must be seen in light of some limitations. Considering that the cause of acute kidney injury is multifactorial, future studies may confirm the contribution of mechanical kidney damage in the increase of blood markers related to AKI. A global analysis of heat strain, metabolic responses, and dehydration should be made to explore the role of kidney mechanical trauma on AKI. The link of impacts assessed in the periphery of the body and mechanical trauma of hard connective tissues must be confirmed in future studies.

Also, it must be explored how much does prolonged massive g-forces impact runners during rough running (e.g., downhill, off-road, mountain) and produce kidney damage compared to similar heavy muscular exercise, but without the massive g-forces. Consequently, it should be explored if downhill running, sudden change of direction, falls, or other similar high magnitude actions produce greater damage than other running actions (e.g., uphill and flat running). Finally, there is a need to use other blood markers (e.g., Cystatin-C, NGAL, KIM-1) that allow researchers to differentiate AKI´s and extreme muscular exercise´s signs and symptoms. There is a need to review AKI′s diagnosis criteria and its validity when applying it to sport sciences and medicine.

## 5. Conclusions

The results suggest that the magnitude and volume of running g-forces monitored with a MARG sensor attached to the lower back of off-road runners could predict the 24% change of serum creatinine and 23% change in serum albumin. These results must be confirmed in future research comparing similar heavy exercise with lower shock loads to the back and kidneys. Although these results may appear promising regarding the potential use of wearable devices to monitor cumulative mechanical kidney trauma in the future, greater understanding is required in the interaction of internal load (e.g., physiological responses) and external load (e.g., accelerations, impacts, decelerations) during prolonged exposure to vigorous repetitive exercise.

The results suggest that a decrease in the amount and magnitude of impacts throughout a session or between sessions can be a way to mitigate the possible collateral damage of acute kidney damage during off-road running. The foregoing considers, therefore, that the monitoring and control of training external and internal loads is essential for the prevention and recovery of AKI in off-road runners. In this sense, it is essential to provide constant feedback on running loads behavior and wearable MARG sensors could be used for these purposes.

## Figures and Tables

**Figure 1 jfmk-05-00093-f001:**
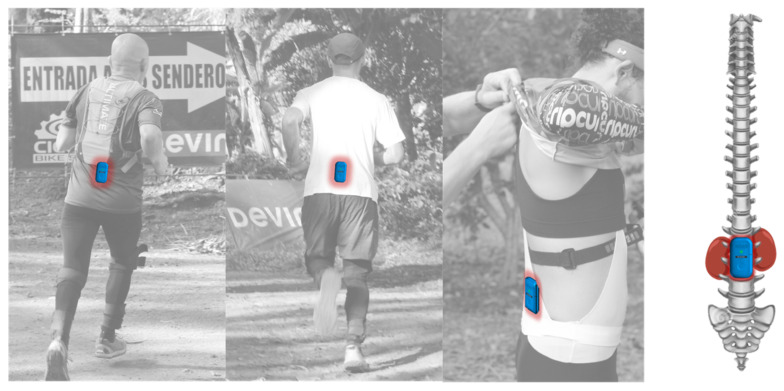
Inertial measurement unit attachment at runner´s lower back (L1–L3).

**Figure 2 jfmk-05-00093-f002:**
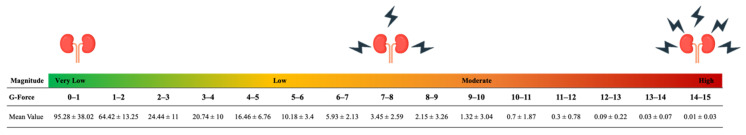
Mean values of impacts per minute associated with 15 g-force categories during off-road mountain running.

**Figure 3 jfmk-05-00093-f003:**
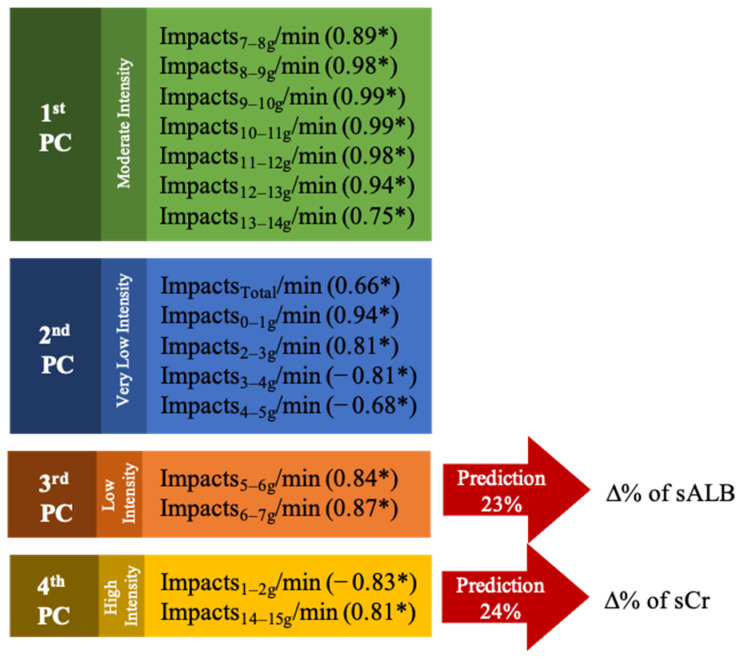
Principal component analysis extracted variables and loadings. * Loadings values.

**Figure 4 jfmk-05-00093-f004:**
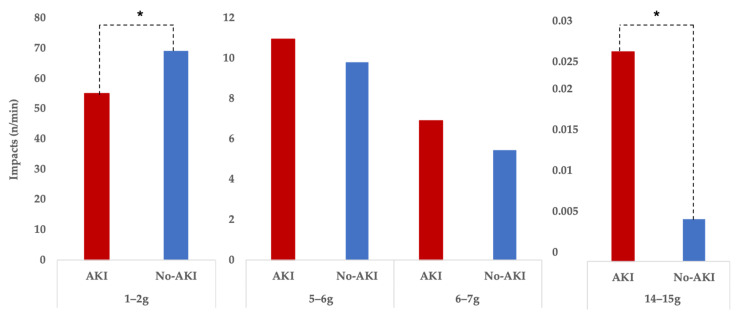
Differences between runners showing signs of AKI (*n* = 6) and those showing no signs of AKI (*n* = 12) regarding impacts per minute, grouped in four impact g-force categories. * The biggest difference between the AKI and no-AKI group is that the no-AKI group managed to run “smoother,” keeping impacts in the lower impact load ranges, while avoiding higher impacts loads.

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
