# Peer review of "Potential Use of Wearable Sensors to Assess Cumulative Kidney Trauma in Endurance Off-Road Running"

_jfmk, 2020, doi:10.3390/jfmk5040093_

Round 1

Reviewer 1 Report

This is a very interesting topic providing evidence for the potential to use non-invasive devices to monitor possible developments of acute kidney injuries. There seems to be sufficient evidence illustrating the effectiveness of MARGs to detect internal changes in the bloodwork. The manuscript is difficult to follow at times, I have included some comments in where I believe the flow is a bit more convoluted. Additionally, there are many spelling and grammatical errors that must be addressed. Regardless, this is an interesting area and I commend your group for a good first draft. I have included a few comments that you may find helpful.

Abstract

  • Line 24: Replace “Pre-race and Post0h” with “pre- and post-race”
  • Line 24: Replace ‘,’ after ‘race’ with ‘:’
  • Line 24: Delete ‘,’ after ‘(sCr)’
  • Line 24-25: Delete ‘were assessed’
  • Line 26: Abbreviate PC after the first time it is mentioned.
  • Line 26-27: Replace ‘principal component’ with PC
  • Line 28: Replace “Pre-race and Post0h” with “pre- and post-race”

Introduction

  • Line 40: Replace ‘levels, increase in serum’ with ‘and’
  • Line 40: Add an ‘s’ to ‘alteration’
  • Line 45: Add a ‘,’ after ‘punches’
  • Line 46: Replace ‘, as endurance running and cycling,’ with ‘(i.e. endurance running and cycling)’
  • Line 46: Replace ‘it’ with ‘AKI’
  • Line 50: Replace ‘special’ with ‘notable’ or something of that nature
  • Line 50: Add ‘which’ between ‘factors’ and ‘may’
  • Line 50: Delete ‘which included’
  • Line 52: Delete ‘a’
  • Line 52: Add ‘s’ to ‘factor’
  • Lines 49-58: This paragraph is convoluted. You begin by describing running and off-road running, which if taking the flow from the end of the previous paragraph, I would assume you would continue with the impact damage. But you talk about dehydration, heat, and metabolic activity, none of which are unique to off-road running. In other words, please make this paragraph flow better.
  • Line 59: Add a ‘,’ after ‘sports’ and ‘running’
  • Line 59: Replace ‘it’ with ’the’

Methods

  • Line 73: I am confused as to why the authors chose to use "pre" and "post0h" as abbreviations for pre- and post-test. These do not need to be abbreviated due to the intuitive nature of the terms. Additionally, no other data were collected at time points other than immediately post-test, correct?

Discussion

  • Line 154: Why 4th and 3rd when the other way around is more logical?
  • Line 158: ureteropelvic
  • Line 162: Replace ‘this’ with ‘these’
  • Line 164: Replace ‘this’ with ‘these’
  • Line 166: Replace ‘have’ with ‘has’
  • Line 180: Delete ‘real’
  • Line 181: Delete ‘may or not’
  • Line 182: Replace ‘anyhow’ with ‘regardless’ or something of that nature
  • Line 182-183: Although this is true, the study conducted was on off-road running. While you can monitor potential for AKI, what preventative strategies can be used to mitigate the extent of mechanical impacts. Additionally, the authors state at the start of this paragraph that AKI may not contribute to chronic renal conditions, but then go on to urge the need to ensure athletes do not have AKI. The message is convoluted.

Author Response

Dear Editor and reviewers:

We have carefully considered all reviewers' recommendations of the paper ID (JFMK-1010081) entitled: Potential use of wearable sensors to assess kidney cumulative trauma in endurance off-road running Please find enclosed our detailed answers to reviewers' queries. The authors declare that the manuscript is original and has not been considered for publication elsewhere. Additionally, the authors had approved the paper for release and are in agreement with its content.

Please find all corrections in ¨track change¨ mode inside the manuscript. The manuscript was corrected following the reviewer´s recommendations and the final version was significantly improved. The manuscript was corrected according to all reviewers suggestions. The scientific evidence supporting the ideas of the paper was reinforced. We really appreciate the effort to improve this final outcome.

Reviewer  1

R1.1. This is a very interesting topic providing evidence for the potential to use non-invasive devices to monitor possible developments of acute kidney injuries. There seems to be sufficient evidence illustrating the effectiveness of MARGs to detect internal changes in the bloodwork. The manuscript is difficult to follow at times, I have included some comments in where I believe the flow is a bit more convoluted. Additionally, there are many spelling and grammatical errors that must be addressed. Regardless, this is an interesting area and I commend your group for a good first draft. I have included a few comments that you may find helpful.

R/ We welcome all the recommendations and errors marked in this review. We definitely trust that the final version of this manuscript has been improved thanks to your input.

R1.2. Abstract

  • Line 24: Replace “Pre-race and Post0h” with “pre- and post-race”
  • Line 24: Replace ‘,’ after ‘race’ with ‘:’
  • Line 24: Delete ‘,’ after ‘(sCr)’
  • Line 24-25: Delete ‘were assessed’
  • Line 26: Abbreviate PC after the first time it is mentioned.
  • Line 26-27: Replace ‘principal component’ with PC
  • Line 28: Replace “Pre-race and Post0h” with “pre- and post-race”

R/ Thank you very much for identifying these errors, all the corrections were made using Track Changes mode.

R1.3. Introduction

  • Line 40: Replace ‘levels, increase in serum’ with ‘and’
  • Line 40: Add an ‘s’ to ‘alteration’
  • Line 45: Add a ‘,’ after ‘punches’
  • Line 46: Replace ‘, as endurance running and cycling,’ with ‘(i.e. endurance running and cycling)’
  • Line 46: Replace ‘it’ with ‘AKI’
  • Line 50: Replace ‘special’ with ‘notable’ or something of that nature
  • Line 50: Add ‘which’ between ‘factors’ and ‘may’
  • Line 50: Delete ‘which included’
  • Line 52: Delete ‘a’
  • Line 52: Add ‘s’ to ‘factor’
  • Line 59: Add a ‘,’ after ‘sports’ and ‘running’
  • Line 59: Replace ‘it’ with ’the’

R/ Thank you very much for highlight the mistakes. All suggested modifications were incorporated into the text.

R1.4. Lines 49-58: This paragraph is convoluted. You begin by describing running and off-road running, which if taking the flow from the end of the previous paragraph, I would assume you would continue with the impact damage. But you talk about dehydration, heat, and metabolic activity, none of which are unique to off-road running. In other words, please make this paragraph flow better.

R/ We really agree in this issue. Authors correct the flow of the paragraph to make it clearer; evidencing that external load is crucial in the development of AKI.

R1.5. Methods

Line 73: I am confused as to why the authors chose to use "pre" and "post0h" as abbreviations for pre- and post-test. These do not need to be abbreviated due to the intuitive nature of the terms. Additionally, no other data were collected at time points other than immediately post-test, correct?

R/The reviewer´s appreciation is corrected. The authors change the terms as suggested throughout the manuscript.

R1.6. Discussion

  • Line 158: ureteropelvic
  • Line 162: Replace ‘this’ with ‘these’
  • Line 164: Replace ‘this’ with ‘these’
  • Line 166: Replace ‘have’ with ‘has’
  • Line 180: Delete ‘real’
  • Line 181: Delete ‘may or not’
  • Line 182: Replace ‘anyhow’ with ‘regardless’ or something of that nature

R/ We appreciate rescuing these errors and providing your recommendations, which were all followed and corrected in the text in change control mode.

R1.7. Line 154: Why 4th and 3rd when the other way around is more logical?

R/ This statistical technique (PCA) is used to analyze those variables of impacts that can be grouped because when correlated they explain a series of data in a certain percentage and it is carried out in an isolated way to the biochemical factors of AKI. These results may be influenced by the quantity, number and distribution of the impacts in each category of G-forces. Regardless of the position of the PC, the interesting thing is to explore which group of these impacts jointly explains the data series and in this case to relate it to the changes in an external variable through regressions.

R1.8. Line 182-183: Although this is true, the study conducted was on off-road running. While you can monitor potential for AKI, what preventative strategies can be used to mitigate the extent of mechanical impacts. Additionally, the authors state at the start of this paragraph that AKI may not contribute to chronic renal conditions, but then go on to urge the need to ensure athletes do not have AKI. The message is convoluted.

R/ Thank you for the opportunity to clarify, the paragraph was rewritten. Although they are two different kidney conditions and the link between AKI and CKD has not been demonstrated; There is enough evidence to suggest that athletes, coaches, and sports scientists should be concerned with controlling the kidney health of runners, monitoring those variables that can trigger AKI, and thus preventing potential cases of this transitory kidney condition.

Reviewer  2

R2.1. The work is original and shows an interesting application for wearable sensors in connection to how kidneys can be challenged during heavy exercise. It looks too straight forward to call the changes in creatinine and albumin Acute Kidney Injury.

R/ Thank you for your appreciations, both the discussion and the conclusions were rewritten in order to avoid absolute interpretations of the results and considering that the etiology of the blood changes may be of varied etiologies as muscle damage. Despite this, the results of the present study should be confirmed in future research, because AKI is a reality in endurance sports and a way to quantify and control it must be found. And from a standpoint of physical burden and trauma damage to the kidneys, this first approach looks promising.

R.2.2. A suggestion for the author is to read articles like False Estimates of Elevated Creatinine Manpreet Samra, MD; Antoine C Abcar, MD Perm J 2012 Spring;16(2):51-52. But with some more careful wording, the manuscript could well be adapted and still shows nice content. It would be incredibly nice if the authors would have the possibility to identify runners that ran relatively smoothly (so with less heavy g-force impacts) and those that ran more "bumpy". In that way one could check for differences in the trends of both markers. The heavy exercise itself would normally be expected to also being capable of temporary shifts in these markers without necessarily kidney damage being present. But note: I am not a Medical Doctor, so an MD review prevails over mine.

R/ On behalf of the authors, we deeply appreciate the observations and recommendations made in this regard. The authors have reviewed the evidence provided by the reviewer and have taken it into account for the analysis of the results of this study, this is reflected in the discussion and conclusions (Please see changes inside the manuscript).

R/ Although there is still discussion about the use of sCr to diagnose AKI during exercise, due to known functional changes at the renal level, elevations of sALB did suggest subclinical damage, this was rescued in the text and noted accordingly.

R/Additionally, to identify if there are differences in the amount of impacts by category. Two groups were made: the first met the AKI criteria and the second did not. Once this was done, the amount of impacts made in each magnitude identified in the PCA was compared.

R2.3. PDF commentaries:

R/ All recommendations made in the PDF document was also corrected in the final version of the manuscript.

Reviewer 2 Report

The work is original and shows an interesting application for wearable sensors in connection to how kidneys can be challenged during heavy exercise. It looks too straight forward to call the changes in creatinine and albumin Acute Kidney Injury. A suggestion for the author is to read articles like False Estimates of Elevated Creatinine Manpreet Samra, MD; Antoine C Abcar, MD Perm J 2012 Spring;16(2):51-52. But with some more careful wording, the manuscript could well be adapted and still shows nice content. It would be incredibly nice if the authors would have the possibility to identify runners that ran relatively smoothly (so with less heavy g-force impacts) and those that ran more "bumpy". In that way one could check for differences in the trends of both markers. The heavy exercise itself would normally be expected to also being capable of temprary shifts in these markers without necessarily kidney damage being present.

But note: I am not a Medical Doctor, so an MD review prevails over mine.

Author Response

(The authors gave the same response as above.)

Round 2

Reviewer 1 Report

Dear authors,

Thank you for the diligent revisions of the manuscript. All my concerns have been addressed.

Author Response

Dear Editor and reviewers:

We have carefully considered all reviewers' recommendations of the paper ID (JFMK-1010081) entitled: Potential use of wearable sensors to assess kidney cumulative trauma in endurance off-road running. Please find enclosed our detailed answers to reviewers' queries. The authors declare that the manuscript is original and has not been considered for publication elsewhere. Additionally, the authors had approved the paper for release and are in agreement with its content.

Please find all corrections in ¨track change¨ mode inside the manuscript. The manuscript was corrected following the reviewer´s recommendations and the final version was significantly improved. The manuscript was corrected according to all reviewers suggestions. The scientific evidence supporting the ideas of the paper was reinforced. We really appreciate the effort to improve this final outcome.

Reviewer  1

R1.1. Dear authors,

Thank you for the diligent revisions of the manuscript. All my concerns have been addressed.

R/ We really appreciate all your recommendation in the first round of the manuscript. This commentaries improved the final version of the study.

Reviewer  2

R2.1. The authors did a very good job in rewriting indeed! Their work to discern between runners with and without signs of AKI increases the value of the article substantially, which is very nicely graphically summarized in figure 4 by showing the differences between the 2 groups of AKI and no AKI.

R/We are very pleased about your commentaries, it really improved the quality of the paper.

R2.2. Note that in Fig 4 the categories 1-2g and 14-15g are both marked with an asterisk (*) but no connected comment is added in the figure caption text. Suggest to add such a text in the caption, maybe like:
* The biggest difference between the AKI and no-AKI group is that the no-AKI group managed to run "smoother", keeping impacts in the lower impact load ranges, while avoiding higher impacts loads.

R/Thank you for your recommendation. The asterisk was explained in the figure caption as requested.

R2.3. In Fig 4, the y-axis now states impacts (n) but given the numbers, that seems strange. Is this maybe impacts per minute?

R/The y-axis was added in order to clarify.

R2.3. For Fig 4, the main caption text could start with: "Differences between runners showing signs of AKI (n= ???) and those showing no signs of AKI (n= ???) regarding impacts per minute???, grouped in 4 impact g-force categories." Then followed by the explanation text about the asterisk marks.

R/Thank you for highlight these issues. We have corrected the caption of Fig 4.

R2.4. Again I have annotated the manuscript with "yellow sticky note" remarks.

R/ All recommendations made in the PDF document was also corrected in the final version of the manuscript.

Reviewer 2 Report

The authors did a very good job in rewriting indeed! Their work to discern between runners with and without signs of AKI increases the value of the article substantially, which is very nicely graphically summarized in figure 4 by showing the differences between the 2 groups of AKI and no AKI. Note that in Fig 4 the categories 1-2g and 14-15g are both marked with an asterisk (*) but no connected comment is added in the figure caption text. Suggest to add such a text in the caption, maybe like:
* The biggest difference between the AKI and no-AKI group is that the no-AKI group managed to run "smoother", keeping impacts in the lower impact load ranges, while avoiding higher impacts loads.

In Fig 4, the y-axis now states impacts (n) but given the numbers, that seems strange. Is this maybe impacts per minute ?

For Fig 4, the main caption text could start with: "Differences between runners showing signs of AKI (n= ???) and those showing no signs of AKI (n= ???) regarding impacts per minute???, grouped in 4 impact g-force categories." Then followed by the explanation text about the asterisk marks.

Again I have annotated the manuscript with "yellow sticky note" remarks.

Author Response

(The authors gave the same response as above.)
